# Early postnatal interactions between beige adipocytes and sympathetic neurites regulate innervation of subcutaneous fat

Jingyi Chi[1], Zeran Lin[1†], William Barr[1†], Audrey Crane[1‡§], Xiphias Ge Zhu[2], Paul Cohen[1*]

[1]Laboratory of Molecular Metabolism, The Rockefeller University, New York, United States; [2]Laboratory of Metabolic Regulation and Genetics, The Rockefeller University, New York, United States

*For correspondence:
pcohen@rockefeller.edu

†These authors contributed equally to this work

Present address: ‡Tufts University School of Medicine, Boston, United States; §Maine Medical Center, Portland, United States

Competing interests: The authors declare that no competing interests exist.

**Abstract** While beige adipocytes have been found to associate with dense sympathetic neurites in mouse inguinal subcutaneous white fat (iWAT), little is known about when and how this patterning is established. Here, we applied whole-tissue imaging to examine the development of sympathetic innervation in iWAT. We found that parenchymal neurites actively grow between postnatal day 6 (P6) and P28, overlapping with early postnatal beige adipogenesis. Constitutive deletion of *Prdm16* in adipocytes led to a significant reduction in early postnatal beige adipocytes and sympathetic density within this window. Using an inducible, adipocyte-specific *Prdm16* knockout model, we found that *Prdm16* is required for guiding sympathetic growth during early development. Deleting *Prdm16* in adult animals, however, did not affect sympathetic structure in iWAT. Together, these findings highlight that beige adipocyte-sympathetic neurite communication is crucial to establish sympathetic structure during the early postnatal period but may be dispensable for its maintenance in mature animals.

## Introduction

The development of adipose tissue was a critical adaptation for our ancestors. White adipose tissue enables the safe storage and rapid mobilization of energy in response to nutritional needs, while brown adipose tissue defends body temperature by dissipating energy as heat (*Rosen and Spiegelman, 2014*). In modern times, however, excess high-calorie foods, a sedentary lifestyle, and thermal comfort have contributed to an overexpansion of white fat and a relative paucity of brown fat (*Heymsfield and Wadden, 2017*; *van Marken Lichtenbelt et al., 2018*). This has resulted in a significant increase in the prevalence of obesity and associated diseases including type 2 diabetes, hypertension, cardiovascular disease, and many types of cancer (*Kopelman, 2000*). Obesity now affects over 40% of adults in the United States and over 600 million adults worldwide (*CDC, 2020a*; *The GBD 2015 Obesity Collaborators, 2017*). Excess adiposity is at the center of the leading causes of morbidity and mortality, and obesity-related medical care costs the United States health care system nearly $150 billion each year (*CDC, 2020b*). Addressing this public health emergency will therefore require new approaches based on a deeper understanding of the tissues and pathways involved in energy homeostasis.

The crucial role of adipose tissue in energy balance has driven great interest in investigating this tissue as a target for the treatment of obesity. While white adipocytes store excess energy, thermogenic brown and beige adipocytes convert lipids and glucose into heat, thereby increasing energy expenditure (*Rosen and Spiegelman, 2014*). Unlike classical brown adipocytes which are

**eLife digest** Mammals have two types of fatty tissue: white fat that mainly stores energy, and brown and beige fat, also known as thermogenic fat, which burns energy to generate heat. In humans, brown fat is associated with potent anti-obesity and anti-diabetes effects. A better understanding of how this type of fat develops and functions could lead to therapeutic strategies to treat these conditions.

Adult human brown fat is similar to rodent inducible brown fat, also known as beige fat. In adult mice, beige fat cells need stimulation from the environment to form. Cold can lead to the generation of beige fat cells by activating a part of the nervous system known as the sympathetic nervous system. In order for this cold-induced formation of beige fat cells to take place, nerves from the sympathetic nervous system must first innervate the fatty tissue. Beige fat cells themselves are important for establishing this innervation, but it was not well understood when and how this occurs.

To study the role of beige fat cells in the establishment of nerve innervation, Chi et al. used genetically modified mice whose beige fat cells are removed when they are treated with an antibiotic called doxycycline. If mice that had not been genetically modified were treated with doxycycline, they developed beige fat cells soon after birth, and these cells shortly became densely innervated by the sympathetic nervous system. However, if the mutant mice were treated with doxycycline around birth, these mice could not make beige fat cells during the treatment and failed to develop dense innervation even when they grew older. These results showed that beige fat cells that form soon after birth are necessary to establish sympathetic nervous system innervation. But are beige fat cells required to maintain this innervation as the mice grow older? To test this, Chi et al. removed them after the innervation was fully established. These mice maintained their innervation, showing that beige fat cells appear to only be required during the establishment of innervation.

Understanding how the sympathetic nervous system establishes its connection to fat so cold can stimulate beige fat formation is a first step to finding new treatments for conditions such as diabetes or obesity. Exploring the timing that underlies the interactions between the sympathetic nervous system and beige fat cells may provide therapeutic targets in this direction.

thermogenic in basal conditions, murine beige adipocytes, which resemble human brown adipocytes in their molecular signature (*Shinoda et al., 2015*), reside in white adipose depots and need to be activated by external stimuli such as the sympathetic nervous system to drive thermogenesis (*Rosen and Spiegelman, 2014*; *Wang and Seale, 2016*). Recent studies have shown that activation of thermogenic adipocytes in both rodents and humans is associated with increased whole-body energy expenditure, improved glucose homeostasis, and enhanced insulin sensitivity (*Becher et al., 2021*; *Cypess et al., 2009*; *Lee et al., 2014*; *Seale et al., 2011*; *van Marken Lichtenbelt et al., 2009*; *Virtanen et al., 2009*), suggesting a new approach to defend against obesity.

The sympathetic nervous system plays a key role in enhancing thermogenic function of brown and beige adipocytes. Although located in distinct fat depots, both brown and beige adipocytes are surrounded by dense sympathetic neurites, termed parenchymal innervation (*Blaszkiewicz et al., 2019*; *Chi et al., 2018b*; *Guilherme et al., 2019*; *Jiang et al., 2017*; *Wirsen, 1964*). Norepinephrine, a neurotransmitter released by these parenchymal neurites, activates β-adrenergic signaling in thermogenic adipocytes, resulting in enhanced thermogenesis and lipolysis (*Cannon and Nedergaard, 2004*; *Hsieh and Carlson, 1957*). The important role of sympathetic stimulation in thermogenesis has driven great interest in understanding the structural and molecular details of sympathetic control of thermogenic adipocytes. Adipocyte-derived factors have been shown to act on the sympathetic nervous system to regulate its structure and activity. Recent studies have identified S100B and TGFβ1 in brown adipocytes as important molecular determinants of sympathetic innervation in brown fat (*Hu et al., 2020*; *Zeng et al., 2019*).

However, it remains largely unclear how beige adipocytes, which are embedded in white fat depots, modulate their sympathetic innervation. Assisted by a whole-adipose immunolabeling and clearing method, called Adipo-Clear, we recently found that the density of sympathetic parenchymal neurites in close apposition to beige adipocytes is dependent on PRDM16 (PR domain containing 16), the transcriptional determinant of beige adipocyte identity and function (*Chi et al., 2018b*).

Specifically, deletion of *Prdm16* in adipocytes led to ablation of beige adipocyte function and dramatically reduced parenchymal innervation density, suggesting that beige adipocyte-associated factors regulate the structure of sympathetic innervation. As neural projections and circuits can be regulated during development and by physiological stimuli in adult animals (*Glebova and Ginty, 2005*; *Holtmaat and Svoboda, 2009*), it is important to determine when the sympathetic innervation surrounding beige adipocytes is established.

Using 3D whole-tissue imaging, we have begun to decipher the timing of the interactions between sympathetic neurites and beige adipocytes in mouse inguinal subcutaneous white fat (iWAT). We found that sympathetic parenchymal innervation in iWAT actively grows during the early postnatal period. Interestingly, we observed that the establishment of dense parenchymal innervation closely follows the development of early postnatal beige adipocytes. To our surprise, using an inducible, adipocyte-specific *Prdm16* knockout mouse model, we found that *Prdm16* in beige adipocytes is required for sympathetic axon growth during early development, but not necessary for maintaining sympathetic structure in adulthood.

## Results

### Sympathetic innervation of iWAT is established during early postnatal development

To better understand adipocyte-sympathetic neurite interactions, we investigated whether the association between beige adipocytes and dense sympathetic innervation is developmentally determined. We first mapped the developmental timing of the sympathetic nervous system in iWAT using Adipo-Clear coupled with light sheet fluorescent imaging. Given that iWAT undergoes active tissue morphogenesis during late embryonic and early postnatal stages (*Wang et al., 2013*), we first performed whole tissue immunostaining and imaging in iWAT isolated from postnatal day (P) two mice using an antibody targeting tyrosine hydroxylase (TH), a maker for sympathetic fibers, which acts as the rate-limiting enzyme in the catecholamine biosynthesis pathway. At this stage, adipocytes appeared fully vascularized and organized into distinct lobular structures, as shown by the endothelial cell marker PECAM (also known as CD31) (*Figure 1—figure supplement 1A,C and F*), consistent with previous reports (*Hong et al., 2015*). While we could detect TH-positive (TH+) signals resembling nerve fascicles as well as fibers wrapping around large blood vessels, dense parenchymal innervation in close apposition to adipocytes, which was reported in adult iWAT (*Chi et al., 2018b*; *Jiang et al., 2017*), was not obvious at this age (*Figure 1—figure supplement 1A–F*).

At P6, more distinct features of sympathetic innervation in iWAT were observed: (a) travelling in parallel within nerve fascicles and (b) wrapping around main blood vessels in a dense mesh-like morphology (*Figure 1A–C*, *Figure 1—figure supplement 2A–D*). Upon further analyzing the innervation pattern across the entire tissue, we observed that these structures were all interconnected to form a continuous sympathetic network. Specifically, we found several convergence points where TH+ nerve fibers within a nerve fascicle deviated from the bundle and merged with the innervation of the central blood vessel (*Figure 1B–C*, *Figure 1—figure supplement 2A and C*, *Figure 1—video 1*), suggesting that sympathetic fibers leave the nerve fascicle and wrap around the main blood vessel as the first order of innervation. Subsequently, the main blood vessel innervation extended around branching arterioles and venules as the second order of innervation (*Figure 1D*, *Figure 1—figure supplement 2E–F*, *Figure 1—video 1*). Lastly, discrete nerve fibers became apparent at the terminals of the second-order innervation to project into tissue parenchyma, where adipocytes are located. Notably, the majority of these nerve fibers appeared to follow capillaries to arrive in the tissue parenchyma (*Figure 1E–F*, *Figure 1—figure supplement 2G–J*). Although nerve endings were visible in the tissue parenchyma at P6, we did not observe any extensive innervation surrounding adipocytes. In addition, both the dorsolumbar and inguinal regions of iWAT showed similar innervation patterns at this stage (*Figure 1—figure supplement 3A–B*, *Figure 1—video 2*). The results from P2 and P6 samples indicate that the sympathetic axons in iWAT first grow along the vasculature before reaching the tissue parenchyma, consistent with previous findings showing that developing sympathetic axons follow the vasculature to reach their target organs (*Glebova and Ginty, 2005*).

Remarkably, adipocyte-innervating neurites became apparent four days later. In the inguinal region, dense parenchymal neurites surrounding adipocytes were first found at P10, in particular

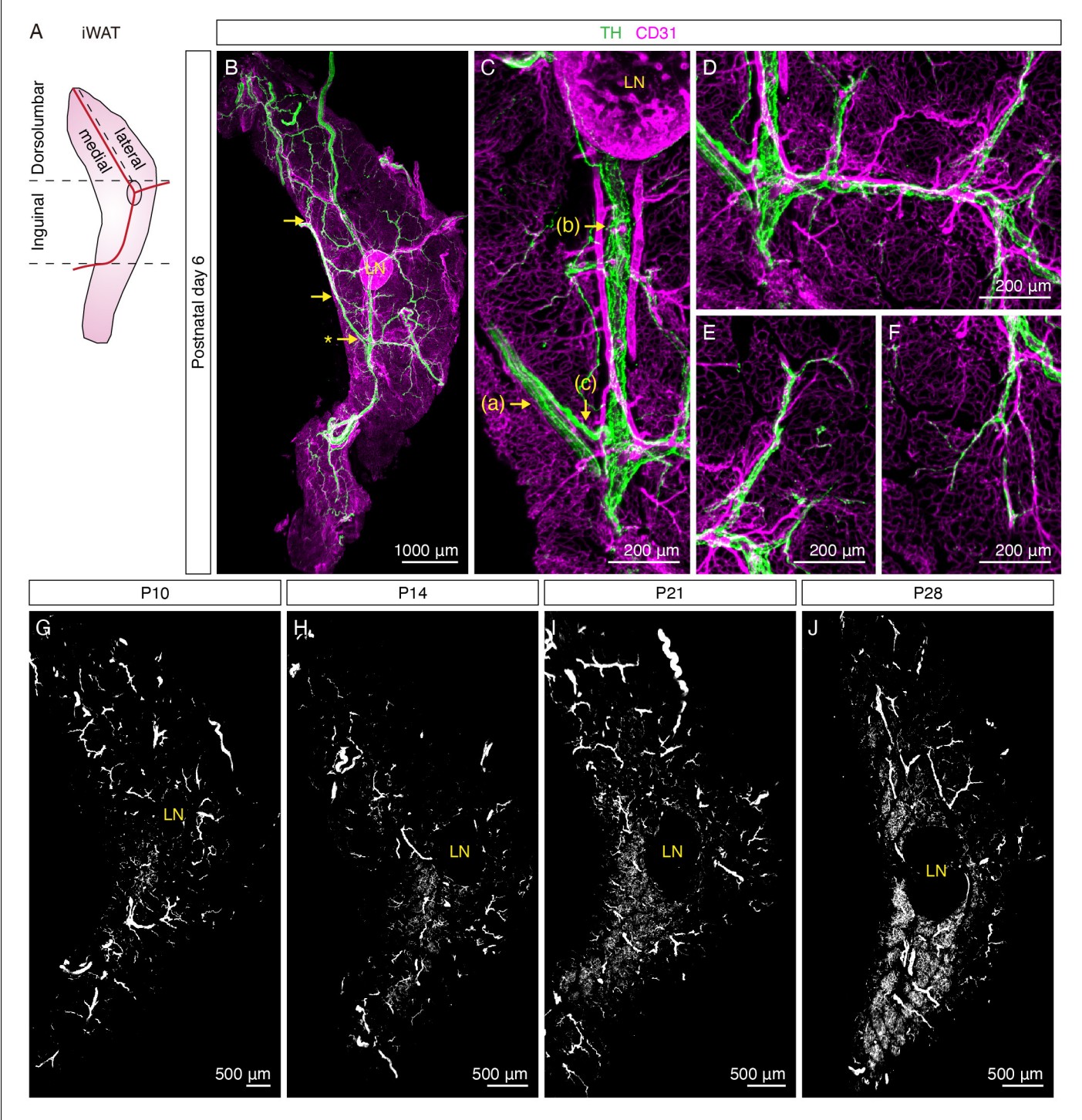

**Figure 1.** Sympathetic innervation of iWAT is established during early postnatal development. (**A**) Schematic view of iWAT. Red lines represent main blood vessels. Using the lymph node and blood vessels as landmarks, iWAT depot is divided into the inguinal and dorsolumbar regions. Dorsolumbar region is further divided into medial and lateral subregions, hereafter referred to as the dorsomedial and dorsolateral regions. Dotted lines indicate boundaries of each region. (**B–F**) Representative images of iWAT from a P6 C57BL6/J mouse immunolabeled with TH (green) and CD31 (magenta). (**B**) Maximum intensity projection (MIP) from a 1000 μm z-stack. Arrows indicate convergence points where nerve fibers deviate from nerve bundles to establish blood vessel innervation. (**C**) High-magnification image of the indicated (*) convergence point in (**B**). Arrows indicate distinct features of sympathetic innervation in iWAT: (**a**) nerve fascicle, (**b**) blood vessel innervation, (**c**) a nerve fiber departing from a nerve fascicle to join blood vessel innervation. (**D**) High-magnification image showing sympathetic innervation from main blood vessel extending to arterioles or venules. (**E–F**) High-magnification images showing discrete nerve endings project into tissue parenchyma. (**G–J**) Representative whole-tissue images of iWAT from (**G**) P10,

*Figure 1 continued on next page*

*Figure 1 continued*

(**H**) P14, (**I**) P21, and (**J**) P28 C57BL6/J mice immunolabeled with TH. MIPs from 50 μm z-stacks are shown. Lymph nodes are indicated as LN. Scale bars are indicated. All imaging studies were performed in at least three independent animals, and representative images are shown.
The online version of this article includes the following video and figure supplement(s) for figure 1:

**Figure supplement 1.** Organization of sympathetic nervous system in iWAT at P2.
**Figure supplement 2.** Organization of sympathetic nervous system in iWAT at P6.
**Figure supplement 3.** Development of sympathetic parenchymal innervation in iWAT.
**Figure 1—video 1.** Corresponding to *Figure 1B–C*.
https://elifesciences.org/articles/64693#fig1video1
**Figure 1—video 2.** Corresponding to *Figure 1—figure supplement 3*.
https://elifesciences.org/articles/64693#fig1video2

within lobules at the core of this region (*Figure 1G*, *Figure 1—figure supplement 2K*, *Figure 1—figure supplement 3D*, *Figure 1—video 2*). At P14, the number of lobules that contain dense parenchymal neurites dramatically increased, spreading outwards from the core of the inguinal region (*Figure 1H*, *Figure 1—figure supplement 3F*, *Figure 1—video 2*). From P21 and onwards, more inguinal lobules were found to harbor dense parenchymal innervation (*Figure 1I–J*, *Figure 1—figure supplement 3H & J*, *Figure 1—video 2*), with the pattern comparable to that of adult iWAT (*Chi et al., 2018b*). Interestingly, the emergence of dense parenchymal neurites in the dorsomedial region lagged behind. While parenchymal neurites were detectable in the dorsomedial region at P14 and P21 (*Figure 1H–I*, *Figure 1—figure supplement 2K–L*, *Figure 1—figure supplement 3E & G*, *Figure 1—video 2*), we did not observe densely innervated lobules that resemble the adult innervation pattern in this region until P28 (*Figure 1J*, *Figure 1—figure supplement 3I*, *Figure 1—video 2*). Notably, the dorsolateral region of iWAT remained sparsely innervated relative to the inguinal region and the dorsomedial region throughout the early postnatal period (*Figure 1G–J*, *Figure 1—figure supplement 2K–L*, *Figure 1—figure supplement 3A,C,E,G & I*, *Figure 1—video 2*).

## UCP1+ beige adipocytes and dense sympathetic parenchymal innervation emerge together during early postnatal development

As our previous findings suggest that beige adipocytes interact with sympathetic projections and modulate the density of sympathetic parenchymal innervation (*Chi et al., 2018b*), we next investigated whether early postnatal development of sympathetic innervation may also be regulated by beige adipocytes. We analyzed the localization of beige adipocytes using an antibody against uncoupling protein 1 (UCP1), a widely accepted marker for thermogenic adipocytes, and compared their distribution in relation to the sympathetic parenchymal innervation in iWAT using whole-tissue imaging. As expected, we observed a strong association between beige adipocytes and parenchymal innervation, even during early postnatal development.

Specifically, we found that beige adipocytes first emerge in iWAT of P6 animals that were born and housed at room temperature, as shown by a few UCP1+ adipocytes sparsely distributed in the core of the inguinal region, close to the inguinal lymph node (*Figure 2A–B*, *Figure 2—figure supplement 1D*). At P10, we detected clusters of UCP1+ adipocytes located in distinct lobules in the core of the inguinal region (*Figure 2C–D*, *Figure 2—figure supplement 1E*). Four days later, at P14, the lobules containing UCP1+ adipocytes further expanded from the core (*Figure 2E & G*, *Figure 2—figure supplement 1F*). At P21 and P28, extensive UCP1+ lobules occupied a significant portion of the inguinal region, comparable to the extent of UCP1+ cells only seen in adult animals after cold exposure (*Figure 2H and J*, *Figure 2—figure supplement 1A,C,G, & H*). On the other hand, the emergence of UCP1+ adipocytes in the dorsolumbar region again lagged behind. UCP1+ adipocytes in the dorsomedial region first emerged in small clusters at P14 and then as distinct lobules at P21 (*Figure 2E–F & and H–I*). At P28, the same region contained a large number of lobules harboring UCP1+ adipocytes (*Figure 2—figure supplement 1A–B*). Interestingly, the dorsolateral region was devoid of UCP1+ adipocytes at all stages analyzed (*Figure 2A,C,E and H*, *Figure 2—figure supplement 1A & D–H*).

To obtain a quantitative measure of these early postnatal beige adipocytes, we also examined mRNA levels of brown and beige adipocyte-enriched genes in the inguinal and dorsolumbar regions

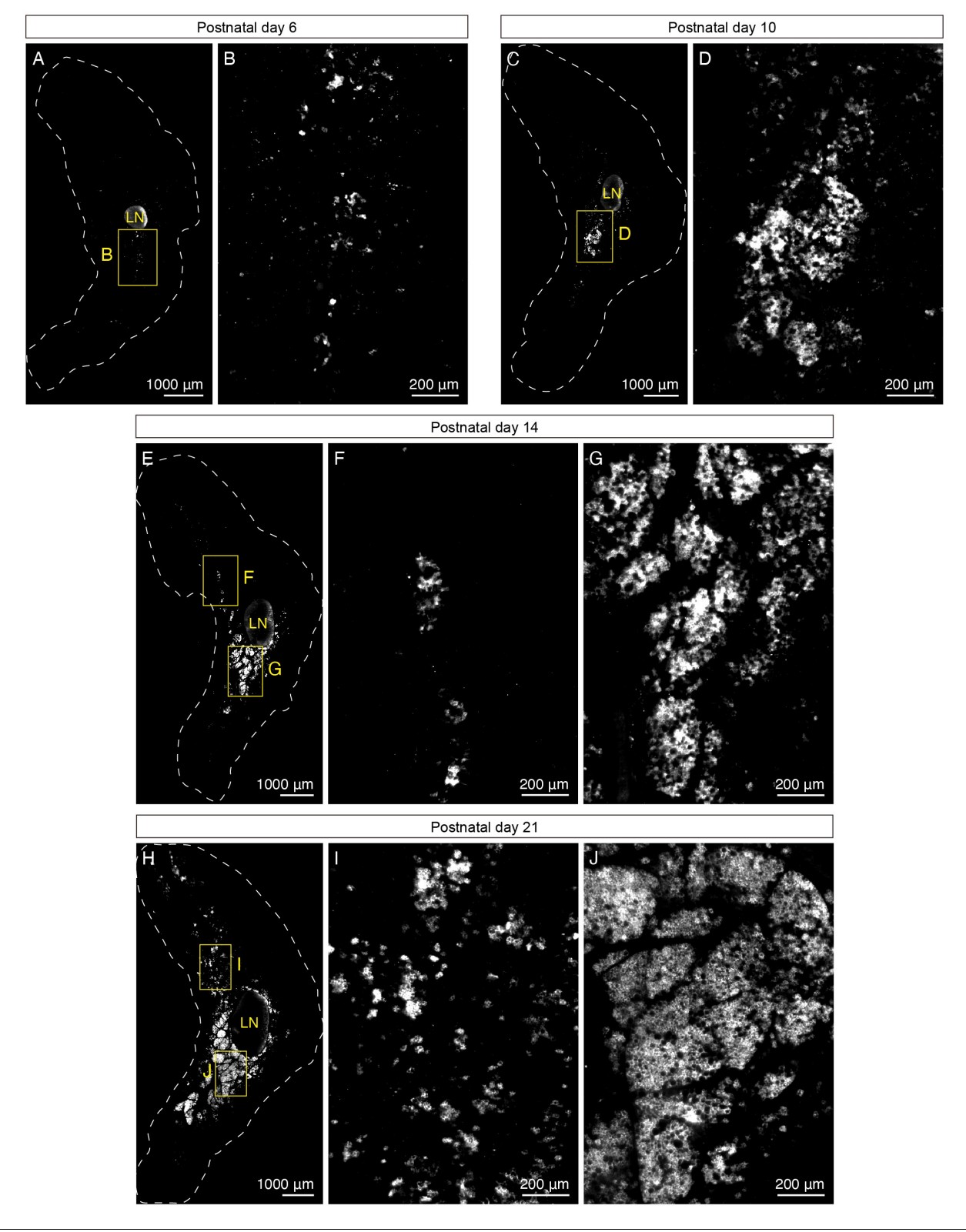

**Figure 2.** UCP1+ beige adipocytes emerge during early postnatal development. (A–J) Representative optical sections of iWAT from P6, P10, P14, and P21 C57BL6/J mice immunolabeled with UCP1. (A) Whole-tissue optical section of a P6 iWAT. (B) High-magnification view of the boxed inguinal region in (A). (C) Whole-tissue optical section of a P10 iWAT. (D) High-magnification view of the boxed inguinal region in (C). (E) Whole-tissue optical section of a P14 iWAT. (F) High-magnification view of the boxed dorsolumbar region in (E). (G) High-magnification view of the boxed inguinal region in (E). (H)

*Figure 2 continued on next page*

*Figure 2 continued*

Whole-tissue optical section of a P21 iWAT. (I) High-magnification view of the boxed dorsolumbar region in (H). (J) High-magnification view of the boxed inguinal region in (H). Lymph nodes are indicated as LN. Dotted lines indicate tissue boundaries based on tissue autofluorescence signals shown in *Figure 2—figure supplement 1D–H*. Scale bars are indicated. All imaging studies were performed in at least three independent animals, and representative images are shown.

The online version of this article includes the following figure supplement(s) for figure 2:

**Figure supplement 1.** UCP1+ beige adipocytes emerge during early postnatal development.

**Figure supplement 2.** UCP1+ beige adipocytes emerge during early postnatal development.

**Figure supplement 3.** UCP1+ beige adipocytes and dense sympathetic parenchymal innervation emerge together during early postnatal development.

of iWAT. In line with the imaging results, *Ucp1* mRNA expression showed a gradual increase from P6 to P14 in the inguinal region (*Figure 2—figure supplement 2A*). To our surprise, although we observed more extensive UCP1+ adipocytes in the inguinal region at P21 and P28 by imaging, *Ucp1* mRNA levels peaked around P12-P16, with the expression level at P14 being fourfold and eightfold higher than that of P21 and P28, respectively (*Figure 2—figure supplement 2A*). Other thermogenic genes (*Cidea* and *Cox8b*) also gradually increased their mRNA expression from P6 to P14, followed by a small downward trend after P21 (*Figure 2—figure supplement 2A*). These quantitative transcriptional data suggest that the beige adipocytes arising in the inguinal region of iWAT during early postnatal development may exhibit peak thermogenic potential around 2–3 weeks of age and gradually become less active as animals mature. On the other hand, all thermogenic genes showed significantly lower mRNA expression in the dorsolumbar region compared to the inguinal region at most time points (*Figure 2—figure supplement 2A*). Of note, *Prdm16*, the transcriptional coregulator that determines beige adipocyte phenotype, showed a consistent 1.5- to twofold increase in mRNA levels in the inguinal relative to the dorsolumbar region across all time points (*Figure 2—figure supplement 2B*). Other transcriptional regulators of beige adipocyte development, such as *Cebpb* and *Ppargc1a*, showed similarly consistent regional differences (around 1.5-fold for *Cebpb* and twofold for *Ppargc1a*) during early development (*Figure 2—figure supplement 2B*). Furthermore, we did not observe any significant regional differences in markers of adipocyte maturation and function (*Fabp4*, *Pparg*, and *Adipoq*) (*Figure 2—figure supplement 2C*). Taken together, these data suggest that adipocytes from the two regions of iWAT are equally differentiated, but the inguinal region may harbor more beige progenitor cells or mature adipocytes with the potential to emerge as beige adipocytes.

When we overlaid the UCP1 and TH signals, we observed a dramatic overlap between the presence of dense parenchymal innervation and beige adipocytes, particularly from P10 onwards (*Figure 2—figure supplement 3A–J*, *Figure 1—video 2*), strongly suggesting that early postnatal beige adipocytes are associated with the signals enabling sympathetic axon growth.

Additionally, as room temperature is considered a mild cold stress to mice, particularly in developing animals that do not have their adult fur pattern, it is possible that early postnatal beige adipocytes arise as a result of cold-induced sympathetic stimulation. When mice were born and raised at a warmer temperature (30°C), at which cold-induced sympathetic firing is minimized, early postnatal beige adipocytes and sympathetic neurites arise in iWAT with the same patterning as that of room temperature-housed mice (*Figure 2—figure supplement 3K–L*). In addition, a recent study using genetic sympathetic ablation showed that early postnatal beige adipocytes develop normally in the absence of sympathetic innervation (*Wu et al., 2020b*). Together, these data suggest that the development of beige adipocytes is likely not dependent on sympathetic activation, but rather based on a developmentally hard-wired program.

## *Prdm16* regulates the emergence of early postnatal beige adipocytes and dense sympathetic parenchymal innervation

We have previously shown that dense parenchymal innervation that localizes to the inguinal region of adult iWAT is significantly reduced by constitutive deletion of *Prdm16* in adipocytes (*Chi et al., 2018b*). To examine whether early postnatal beige adipocytes and their regulation of dense

symbathetic innervation are also dependent on *Prdm16*, we analyzed iWAT of adipocyte-specific *Prdm16* knockout mice (*Adipoq-Cre; Prdm16^{lox/lox}*; hereafter noted as constitutive *Prdm16*^{KO} or c*Prdm16*^{KO} mice) at postnatal days 6, 14, and 21, key time points in the course of beige adipocyte and sympathetic innervation development.

At P6, we observed minimal beige adipocytes and scant parenchymal innervation in both control and c*Prdm16*^{KO} mice (*Figure 3—figure supplement 1A–H*), suggesting that the sympathetic nervous system develops similarly in both models prior to the emergence of beige adipocyte clusters. At P14, the deletion of *Prdm16* completely ablated beige adipocytes that normally arise in the inguinal region of control mice, both at the mRNA and protein levels (*Figure 3I*, *Figure 3—figure supplement 1I*, *Figure 3—figure supplement 2A–D*). Correspondingly, the increase seen in parenchymal innervation density in the inguinal region of control mice was not observed in c*Prdm16*^{KO} mice (*Figure 3A–D*, *Figure 3—figure supplement 2A–D*). At P21, we observed similar ablation of beige adipocytes and lack of growth in parenchymal innervation in c*Prdm16*^{KO} relative to control samples (*Figure 3E–I*, *Figure 3—figure supplement 1I*, *Figure 3—figure supplement 2E–H*). These results indicate that early postnatal beige adipocytes indeed depend on PRDM16, the well-characterized transcriptional determinant of brown and beige adipocytes. Importantly, these data strongly suggest that sympathetic axon growth during early iWAT morphogenesis may be regulated by PRDM16-dependent signals.

Although PRDM16 is known to be important for beige adipocyte function, it remains possible that ablation of *Prdm16* in all adipocytes by *Adipoq*-Cre altered white adipocyte function and therefore affected sympathetic innervation. To address this, we assessed *Prdm16* mRNA and protein levels in the inguinal and dorsolateral regions of iWAT, which are predominantly beige and white regions, respectively (*Figure 2*). We performed qPCR on the two regions isolated from control and c*Prdm16*^{KO} mice at P14 (*Figure 3—figure supplement 2I*). The control dorsolateral region showed significantly lower expression level of *Prdm16* mRNA than the control inguinal region. Importantly, the *Prdm16* mRNA level in the control dorsolateral region was indistinguishable from that in c*Prdm16*^{KO} dorsolateral or inguinal regions, suggesting that the wild-type dorsolateral region naturally expresses very low levels of *Prdm16* mRNA with levels indistinguishable from *Prdm16* knockout samples. We further assessed PRDM16 protein levels across multiple fat depots of young adult mice (*Figure 3—figure supplement 2J*). Consistently, the dorsolateral region exhibited a considerably lower level of PRDM16 compared with the inguinal region in wild-type iWAT, while there were no detectable levels of PRDM16 in the iWAT of c*Prdm16*^{KO} mice or wild-type eWAT. Although there was still a minimal level of PRDM16 protein in the dorsolateral region of iWAT, this may be attributed to the small number of beige adipocytes in this region. Altogether, *Prdm16* appears to be minimally expressed in white adipocytes in iWAT, and thus its deletion in white adipocytes is likely to contribute minimally to the changes in sympathetic innervation.

Interestingly, although PRDM16 also plays a critical role in brown adipocyte determination and function, deletion of *Prdm16* in interscapular brown fat (iBAT) does not affect its development or thermogenic function in young adults (*Cohen et al., 2014*; *Harms et al., 2014*). Previous studies have shown that the role of PRDM16 in iBAT formation and function is compensated for by PRDM3, a transcriptional regulator closely related to PRDM16 (*Harms et al., 2014*). Consistent with these findings, we detected similarly extensive sympathetic parenchymal innervation in iBAT of both control and c*Prdm16*^{KO} mice (*Figure 3—figure supplement 2K–L*).

## *Prdm16* deletion during early development causes decreased sympathetic parenchymal innervation

To further delineate the critical time window for sympathetic innervation patterning in iWAT, we generated an inducible *Prdm16* knockout mouse model (*Adipoq^{rtTA}; TRE-Cre; Prdm16^{lox/lox}*; hereafter noted as inducible *Prdm16*^{KO} or i*Prdm16*^{KO} mice), where *Prdm16* can be deleted in adipocytes in a doxycycline-dependent manner (*Figure 4A*). To test whether sympathetic parenchymal innervation may be developmentally determined during a defined time window, doxycycline was delivered to mice from embryonic day (E) 14 until P21, the period of time when both beige adipocytes and parenchymal innervation development become clearly apparent. Following doxycycline treatment, i*Prdm16*^{KO} and littermate control mice were switched back to chow diet for 2 weeks and subsequently exposed to either room temperature (RT) or 8°C for 2 days (*Figure 4B*).

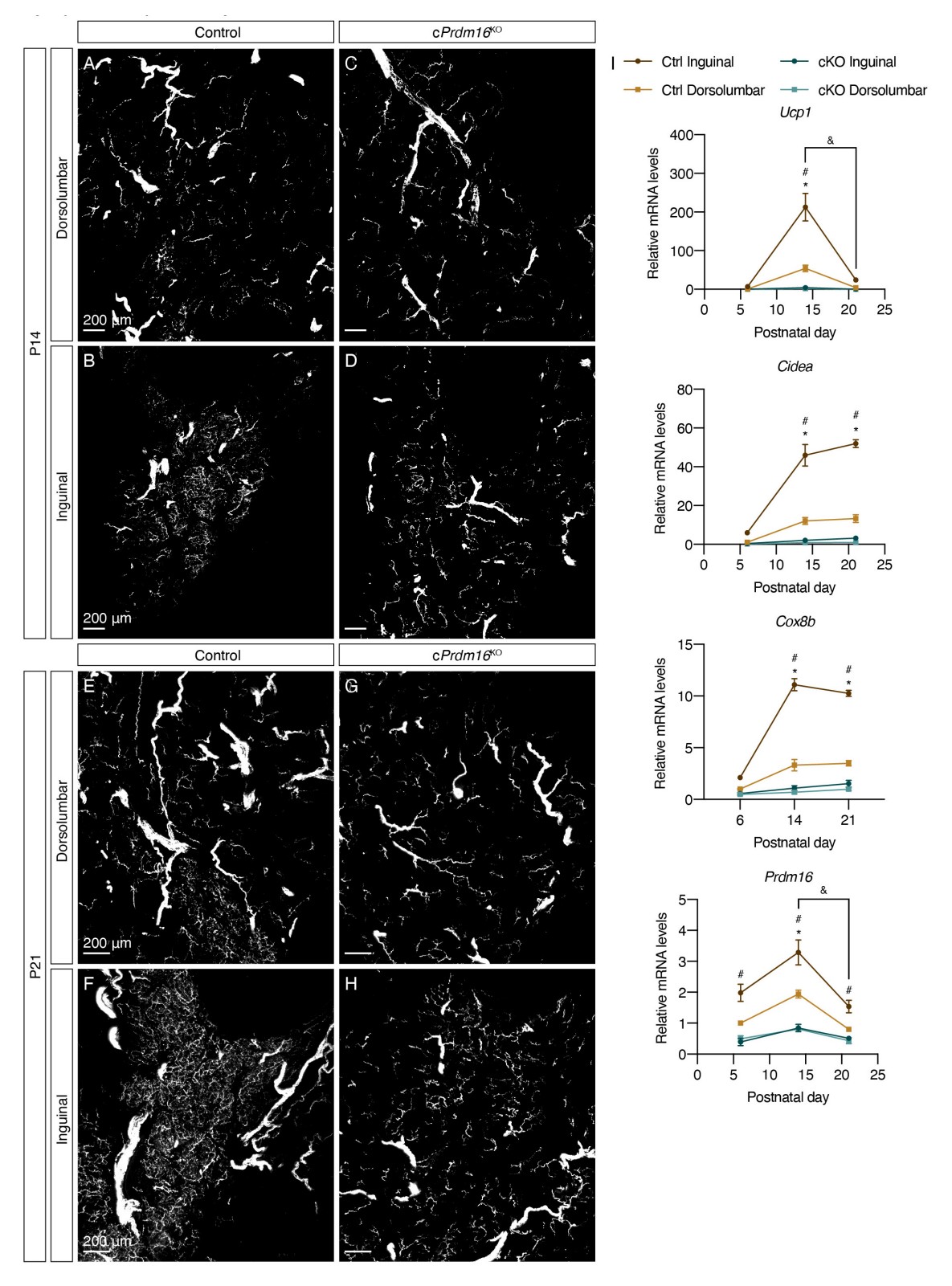

**Figure 3.** *Prdm16* regulates the emergence of early postnatal beige adipocytes and dense sympathetic parenchymal innervation. (A–D) Representative images of iWAT from (A–B) control and (C–D) c*Prdm16*^KO mice at P14, immunolabeled with TH. MIPs from 50 µm z-stacks are shown. (E–H) Representative images of iWAT from (E–F) control and (G–H) c*Prdm16*^KO mice at P21, immunolabeled with TH. MIPs from 50 µm z-stacks are shown. (A, C, E and G) Images of the dorsolumbar region. (B, D, F and H) Images of the inguinal region. Scale bars are indicated. Imaging was performed in at

*Figure 3 continued on next page*

*Figure 3 continued*

least three independent animals per genotype, and representative images are shown. (I) Normalized gene expression of dorsolumbar vs. inguinal regions in iWAT from control and c*Prdm16*^KO mice at P6, P14, and P21, n = 2–3. Representative genes involved in the thermogenic program are shown. Data are presented as mean ± SEM and analyzed by two-way ANOVA followed by Bonferroni's multiple comparisons test. * denotes p<0.05 dorsolumbar vs. inguinal regions of control samples at each time point. # denotes p<0.05 inguinal regions of control vs. c*Prdm16*^KO samples at each time point. & denotes p<0.05 inguinal region at P14 in comparison to inguinal regions at P21.

The online version of this article includes the following figure supplement(s) for figure 3:

**Figure supplement 1.** *Prdm16* regulates the emergence of early postnatal beige adipocytes and dense sympathetic parenchymal innervation.

**Figure supplement 2.** *Prdm16* regulates the emergence of early postnatal beige adipocytes and dense sympathetic parenchymal innervation.

We first confirmed that perinatal doxycycline treatment led to a strong reduction in *Prdm16* in adipose depots at the mRNA (70% reduction in the inguinal region of iWAT, 65% reduction in iBAT) and protein levels (*Figure 4—figure supplement 1A–B*). Both i*Prdm16*^KO (perinatal, 5 weeks) and littermate control mice developed similarly, with no difference in body mass (*Figure 4—figure supplement 1C*). To examine whether the perinatal deletion of *Prdm16* led to changes in downstream thermogenic gene expression, we analyzed mRNA levels of thermogenic markers that are induced by cold in wild-type iWAT. Since we have consistently observed that the inguinal region of iWAT is enriched in both early postnatal and cold-inducible beige adipocytes (*Chi et al., 2018b*), we hereafter focused on analyzing changes in the inguinal region. As expected, the control inguinal regions exhibited robust increases in the entire panel of thermogenic markers following 2 days of cold exposure (e.g. 35-fold induction in *Ucp1*) (*Figure 4C*, *Figure 4—figure supplement 1D*). In contrast, the cold-induced increases were almost completely blocked following perinatal deletion of *Prdm16* (*Figure 4C*, *Figure 4—figure supplement 1D*), mirroring our previous findings in constitutive *Prdm16*^KO mice (*Cohen et al., 2014*). On the other hand, the transient *Prdm16* deletion did not significantly alter adipocyte differentiation or function, as shown by *Pparg*, *Fabp4*, and *Adipoq* mRNA levels (*Figure 4—figure supplement 1E*).

We next performed whole-tissue imaging on cleared iWAT of control and i*Prdm16*^KO to assess whether perinatal *Prdm16* deletion is sufficient to reproduce the innervation defect seen in c*Prdm16*^KO. While the gross features of sympathetic innervation such as nerve fascicles and blood vessel innervation were preserved in perinatal i*Prdm16*^KO, the parenchymal innervation that localized to the inguinal region of control iWAT was dramatically reduced in perinatal i*Prdm16*^KO (*Figure 4D–I*, *Figure 4—video 1*). To quantitatively assess differences in parenchymal neurite density from the 3D images, we computationally traced and measured the parenchymal neurite lengths in randomly selected tissue volumes contained within lobules in the inguinal region of iWAT (*Figure 4—figure supplement 2A–D*). When neurite lengths were normalized to volumes of the isolated cuboids, perinatal i*Prdm16*^KO samples showed a 74.5% reduction in parenchymal neurite density compared to control samples (*Figure 4J*, *Figure 4—figure supplement 2F*). Interestingly, we noticed that the adipocytes in perinatal i*Prdm16*^KO samples also appeared larger in size as outlined by the tissue autofluorescence and vasculature signal (*Figure 4—figure supplement 1F–K*). It is possible that the decreased neurite density in i*Prdm16*^KO was due to a simple scaling effect; that is, the neurites appear more sparse because the adipocytes are larger in size. To rule out this possibility, we also calculated neurite density by factoring in adipocyte density from each tissue volume, and still observed a significant 52.8% decrease in density following perinatal *Prdm16* deletion (*Figure 4K*, *Figure 4—figure supplement 2E–F*).

Similar to our previous findings with the c*Prdm16*^KO model, perinatal *Prdm16* knockout in adipocytes did not significantly affect thermogenic or adipogenic markers in iBAT (*Figure 4—figure supplement 3A–B*). In line with this, the sympathetic parenchymal innervation levels did not appear different between control and perinatal i*Prdm16*^KO iBAT samples (*Figure 4—figure supplement 3C–D*).

To assess whether the innervation defect following perinatal deletion of *Prdm16* may be rescued as animals age, we performed the same perinatal deletion of *Prdm16* but let animals reach 8 weeks of age on a chow diet, hereafter referred to as i*Prdm16*^KO (perinatal, 8 weeks) (*Figure 4—figure*

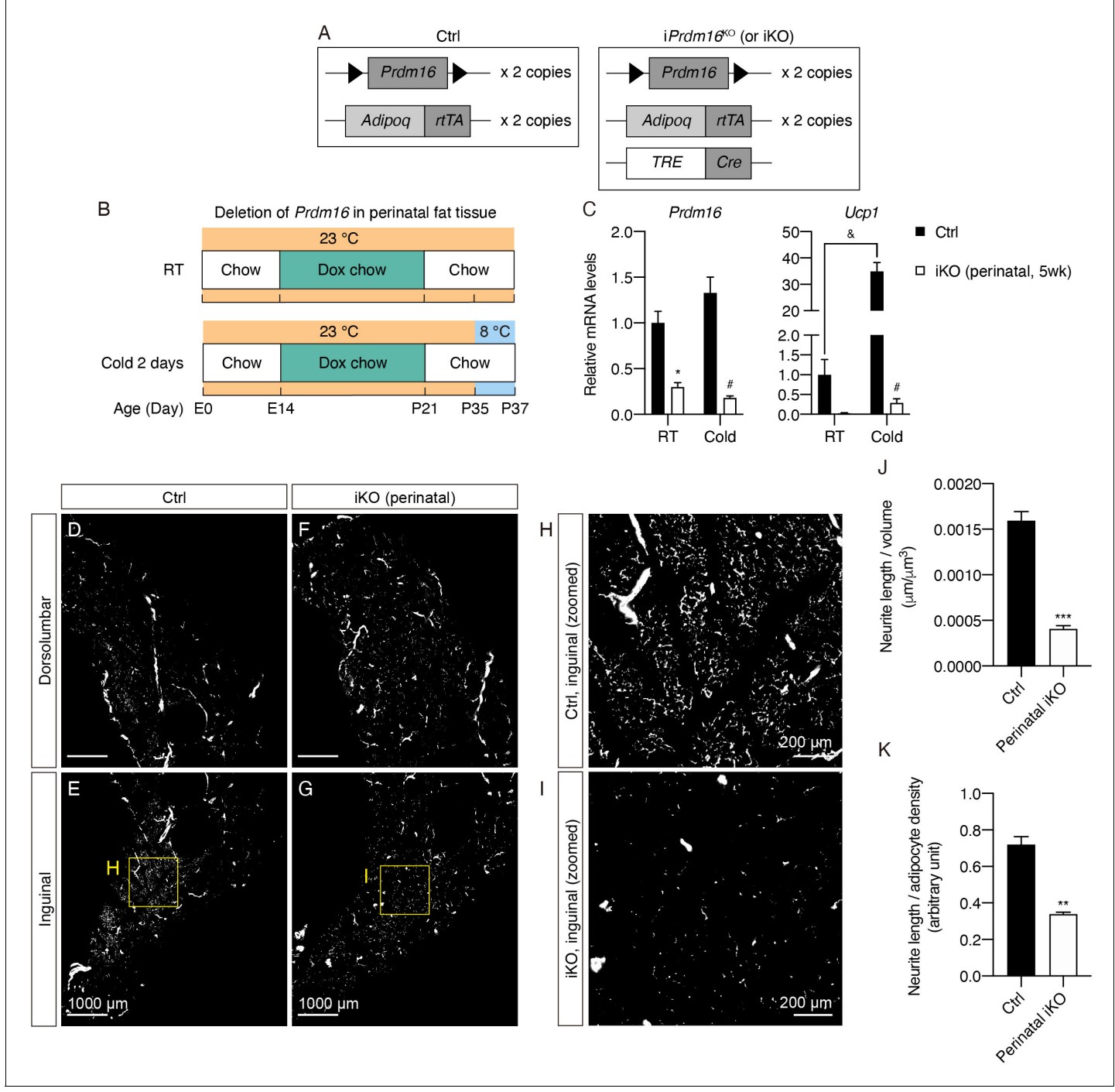

**Figure 4.** *Prdm16* deletion during early development causes decreased sympathetic parenchymal innervation. (**A**) Schematic representation of the genetic components of the control and i*Prdm16*KO mice. i*Prdm16*KO mice carry floxed *Prdm16* alleles (*Prdm16*lox/lox), two copies of *Adipoq*rtTA transgene, and one copy of *TRE-Cre* transgene. Littermates carrying only *Prdm16*lox/lox and *Adipoq*rtTA (i.e. Cre-) were used as the control animals. (**B**) Control (Cre-) and i*Prdm16*KO (Cre+) mice housed at RT (23°C) were kept on a doxycycline-containing chow diet from E14 until P21 before being switched to a regular chow diet for another 2 weeks. Control and i*Prdm16*KO mice were either maintained at RT (23°C) or exposed to cold (8°C) for 2 days at P35. (**C**) Normalized gene expression of inguinal regions from control and i*Prdm16*KO (perinatal, 5 weeks) mice exposed to RT or cold, n = 3–5. Data are presented as mean + SEM and analyzed by two-way ANOVA followed by Bonferroni's multiple comparisons test. * denotes p<0.05 iKO (perinatal, 5 weeks) vs. control samples at RT. # denotes p<0.05 iKO (perinatal, 5 weeks) vs. control samples at cold. & denotes p<0.05 cold- vs. RT-exposed control samples. (**D–I**) Representative optical sections of iWAT from (**D–E**) control and (**F–G**) i*Prdm16*KO (perinatal, 5 weeks) mice maintained at RT, immunolabeled with TH. (**D and F**) Images of the dorsolumbar region. (**E and G**) Images of the inguinal region. (**H**) High-magnification optical section of the boxed region in (**E**). (**I**) High-magnification optical section of the boxed region in (**G**). Scale bars are indicated. (**J and K**) Quantification of

*Figure 4 continued on next page*

*Figure 4 continued*

sympathetic parenchymal innervation in inguinal regions with total neurite length normalized to (J) regional volume or (K) adipocyte density. N = 3 biological replicates per genotype were analyzed. Average neurite density from five to seven randomly selected tissue volumes (technical replicates) contributes to neurite density measurement of one biological sample. Data are presented as mean + SEM and analyzed by Student's t test. ** and *** denote p<0.01 and p<0.001, respectively.

The online version of this article includes the following video and figure supplement(s) for figure 4:

**Figure supplement 1.** *Prdm16* deletion during early development blocks cold-induced thermogenic gene expression and causes decreased sympathetic parenchymal innervation in iWAT.

**Figure supplement 2.** Neurite density quantification in 3D images.

**Figure supplement 3.** *Prdm16* deletion during early development does not affect thermogenic gene program or sympathetic innervation in iBAT.

**Figure supplement 4.** *Prdm16* deletion during early development causes lasting defects in sympathetic parenchymal innervation.

**Figure 4—video 1.** Corresponding to *Figure 4*.

https://elifesciences.org/articles/64693#fig4video1

---

*supplement 4A*). As iWAT continues to expand in young adult mice, we reasoned that newly developed adipocytes during iWAT expansion might also affect sympathetic innervation. Consistent with the results from the iPrdm16$^{KO}$ (perinatal, 5 weeks) experiment, we observed significantly reduced expression of *Prdm16* and other thermogenic genes at the mRNA level in iPrdm16$^{KO}$ (perinatal, 8 weeks) relative to control (*Figure 4—figure supplement 4B–D*). Importantly, TH+ fibers still appeared substantially sparser in iPrdm16$^{KO}$ (perinatal, 8 weeks) than control (*Figure 4—figure supplement 4E–J*). Taken together, these results demonstrate that interactions between beige adipocytes and sympathetic nerve endings during an early critical developmental window are required for establishment of the sympathetic network in iWAT, as perturbations during this window lead to lasting effects on parenchymal innervation density.

## *Prdm16* is not required for maintaining dense sympathetic parenchymal innervation in mature iWAT

Next, we assessed whether PRDM16 is also important for maintaining sympathetic parenchymal innervation in adult animals. To that end, we started doxycycline treatment when mice were 8 weeks of age. Following 4 weeks of doxycycline treatment, iPrdm16$^{KO}$ (adult deletion) and littermate control mice were placed at either RT or 8°C for 2 days to allow subsequent analysis of the thermogenic gene program (*Figure 5A*). As expected, doxycycline treatment in adult mice led to a robust reduction in *Prdm16* at the mRNA (83% in the inguinal region of iWAT at RT) and protein levels (*Figure 5B*, *Figure 5—figure supplement 1A*). No body mass difference was observed between iPrdm16$^{KO}$ (adult deletion) and control mice (*Figure 5—figure supplement 1B*). We also observed significant attenuation of cold-induced upregulation in all thermogenic markers following *Prdm16* deletion (*Figure 5B*, *Figure 5—figure supplement 1C*). Consistent with our previous findings, *Prdm16* deletion did not significantly alter markers of adipocyte differentiation (*Figure 5—figure supplement 1D*).

Interestingly, we found dense sympathetic parenchymal innervation present in the inguinal regions of iWAT from both iPrdm16$^{KO}$ (adult deletion) and control samples (*Figure 5C–H*, *Figure 4—video 1*). Quantitative analysis of the parenchymal neurite density (normalized by volume) from the 3D images resulted in a small, insignificant decrease (17.6%) in neurite density in adult iPrdm16$^{KO}$ relative to control samples (*Figure 5I*). Furthermore, as the adipocytes appeared larger in adult iPrdm16$^{KO}$ samples (*Figure 5—figure supplement 1E–J*), we normalized neurite length by adipocyte density. As a result, we observed a reverse relationship with the neurite density in adult iPrdm16$^{KO}$ being slightly higher (1.23-fold) than that of controls (*Figure 5J*). These results suggest that the small density difference when normalized by volume is likely driven by a scaling effect rather than actual neurite remodeling. Importantly, these data indicate that deleting *Prdm16* in adipocytes of adult animals causes minimal changes in sympathetic parenchymal neurite density, strongly supporting the early postnatal period being the critical window during which interactions between beige adipocytes and sympathetic nerve endings occur.

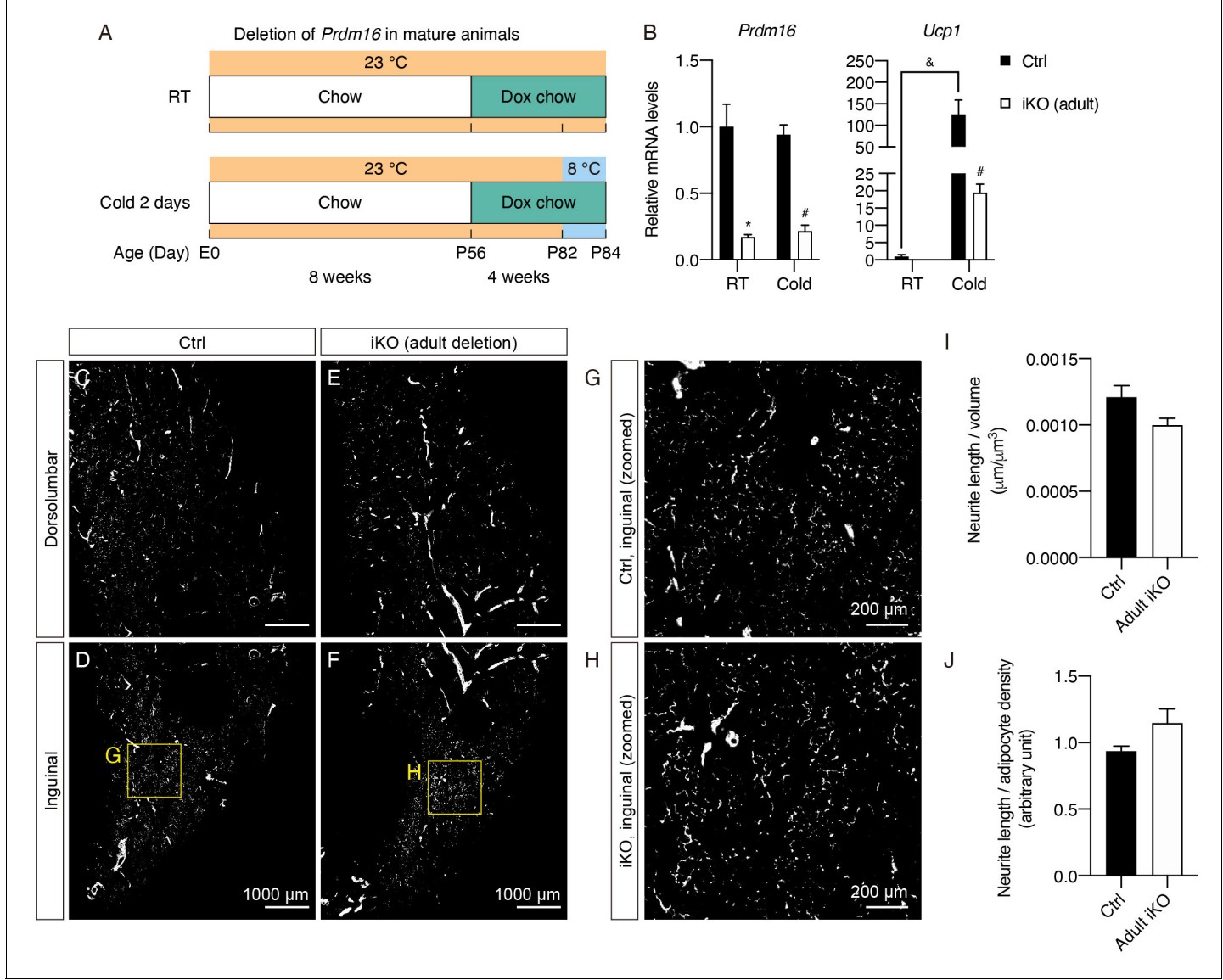

**Figure 5.** *Prdm16* is not required for maintaining sympathetic parenchymal innervation in mature iWAT. (A) Control (Cre-) and i*Prdm16*KO (Cre+) mice housed at RT (23°C) were kept on a regular chow diet until 8 weeks of age before being switched to a doxycycline-containing chow diet for 4 weeks. Control and i*Prdm16*KO mice were either maintained at RT (23°C) or exposed to cold (8°C) for 2 days at the end of doxycycline treatment. (B) Normalized gene expression of inguinal regions from control and i*Prdm16*KO (adult) mice exposed to RT or cold, n = 3–5. Data are normalized to control RT group, presented as mean + SEM, and analyzed by two-way ANOVA followed by Bonferroni's multiple comparisons test. * denotes p<0.05 iKO (adult) vs. control samples at RT. # denotes p<0.05 iKO (adult) vs. control samples at cold. & denotes p<0.05 cold- vs. RT-exposed control samples. (C–F) Representative optical sections of iWAT from (C–D) control and (E–F) i*Prdm16*KO (adult deletion) mice maintained at RT, immunolabeled with TH. (C and E) Images of the dorsolumbar region. (D and F) Images of the inguinal region. (G) High-magnification optical section of the boxed region in (D). (H) High-magnification optical section of the boxed region in (F). Scale bars are indicated. (I–J) Quantification of sympathetic parenchymal innervation in inguinal regions with total neurite length normalized to (I) regional volume or (J) adipocyte density. N = 3 biological replicates per genotype were analyzed. Average neurite density from four to eight randomly selected tissue volumes (technical replicates) contributes to neurite density measurement of one biological sample. * denotes p<0.05 analyzed by Student's t test.

The online version of this article includes the following video and figure supplement(s) for figure 5:

**Figure supplement 1.** *Prdm16* deletion in adulthood attenuates cold-induced thermogenic gene expression but does not affect sympathetic parenchymal innervation in iWAT.

**Figure supplement 2.** Additional tissue structures in the core of the inguinal region.

**Figure 5—video 1.** Corresponding to *Figure 5*.

https://elifesciences.org/articles/64693#fig5video1

## Discussion

Sympathetic innervation plays an important role in regulating two key aspects of adipose tissue function: lipolysis and thermogenesis. Dense sympathetic parenchymal innervation is observed in both iBAT and the inguinal region of iWAT, where beige adipocytes are primarily located; however, sparse innervation is found in eWAT and the dorsolateral region of iWAT, an area of the tissue that is devoid of beige adipocytes even under long term cold stimulation (*Chi et al., 2018b*; *Dichamp et al., 2019*; *Huesing et al., 2020*; *Murano et al., 2009*; *Zeng et al., 2019*). The strong association between thermogenic adipocytes and dense sympathetic neurites as well as the association between white adipocytes and sparse innervation suggests that adipocyte type may determine the density of sympathetic parenchymal innervation. Indeed, when beige-to-white adipocyte identity change was achieved by adipocyte-specific deletion of *Prdm16*, the density of sympathetic parenchymal innervation was significantly reduced in the inguinal region of iWAT (*Chi et al., 2018b*). Although many previous studies have attempted to define the relationships between sympathetic neurite density and beige adipocytes, when and how such relationships are established has remained unclear.

Here, we delineated key stages of sympathetic nervous system development in iWAT using whole tissue imaging. Specifically, we observed that sympathetic parenchymal innervation in close apposition to adipocytes is established between P6 and P28. Importantly, the appearance of UCP1+ beige adipocytes precedes the emergence of dense parenchymal neurites during early postnatal development. We further demonstrated that both early postnatal beige adipocytes and dense parenchymal neurites depend on *Prdm16* expression in adipocytes. Using an inducible *Prdm16* deletion model, we identified an early critical period during which beige adipocytes modulate sympathetic axon growth. However, *Prdm16* deletion in adult mice did not alter the sympathetic structure.

Assisted by whole-tissue images, our study carefully examined the growth of sympathetic axons in iWAT. We observed that sympathetic fibers travel in nerve fascicles to arrive at iWAT, and then depart from nerve fascicles to reach main blood vessels within iWAT. By P6, the sympathetic innervation on blood vessels demonstrated a dense mesh-like structure, resembling that of mature iWAT. However, dense parenchymal innervation surrounding adipocytes, which several studies have characterized in mature iWAT (*Chi et al., 2018b*; *Jiang et al., 2017*), is not established at P6. Instead, we observed sparse discrete sympathetic fibers and small vessels in congruence within the tissue parenchyma, a common phenomenon in sympathetic nerve fiber development where vasculature serves as a guide to direct developing fibers to reach their targets, such as the heart (*Nam et al., 2013*). Subsequently, from P10 until P28, dense parenchymal innervation becomes obvious where clusters of UCP1+ beige adipocytes are located.

Although beige adipocytes emerge in white fat depots of adult mice following cold challenge, we observed strong beige adipogenesis during early postnatal development, similar to the 'peri-weaning' beige adipocytes reported in recent studies (*Wu et al., 2020a*; *Wu et al., 2020b*). Using high-resolution whole tissue imaging focusing on the inguinal region of iWAT, we detected scattered UCP1+ adipocytes at P6, subsequent emergence of small UCP1+ adipocyte clusters at P10, and an expansion to almost all lobules from P14 to P28. On the other hand, beige adipogenesis is delayed in the dorsolumbar region, where small clusters of beige adipocytes are not detected until P14. It will be of interest to investigate the mechanisms driving the preferential localization of early postnatal beige adipocytes. Although beige adipocyte recruitment in adult mice requires sympathetic stimulation, early postnatal beige adipocytes develop normally in mice born and raised at thermoneutrality with minimal sympathetic activity as well as in mice with genetic sympathetic ablation (*Wu et al., 2020b*). Interestingly, we found several genes involved in the transcriptional control of beige adipocyte determination and function to have higher expression levels in the inguinal than the dorsolumbar region throughout early development, suggesting adipocytes or their progenitors in different regions may exhibit unique properties. Additionally, as early postnatal beige adipocytes first emerge close to the core of the inguinal region, other tissue structures such as blood or lymph vessels that are found in the same area (*Figure 1B–C* and *Figure 5—figure supplement 2A–E*) may play a role in promoting early beige adipogenesis.

It is also worth noting that the whole-tissue UCP1+ patterns at P21 and P28 closely resemble that of adult mice following cold exposure. In addition, the thermogenic gene expression of early postnatal beige adipocytes diminishes as mice mature. It is possible that early postnatal beige adipocytes

gradually become inactivated during maturation, and cold challenge re-activates the same cells in adult mice. Follow-up studies will be needed to examine the fate of early postnatal beige adipocytes. Of note, emerging studies have described additional beige adipocytes that rely on pathways other than UCP1 to dissipate heat (*Chen et al., 2019*; *Ikeda et al., 2017*; *Kazak et al., 2015*). Thorough characterization of these beige adipocytes using specific markers will be needed to delineate their development and fate.

Using constitutive and inducible adipocyte-specific *Prdm16* deletion models, we ablated early postnatal beige adipocyte function and found that this dramatically reduced parenchymal sympathetic neurites in iWAT. Our findings suggest that early postnatal beige adipocytes may express PRDM16-dependent neurotrophic factors that stimulate sympathetic axon growth or downregulate inhibition cues. Recent studies have unveiled important roles of brown adipocyte-derived factors, such as S100B and TGFβ1, in regulating sympathetic innervation in iBAT (*Hu et al., 2020*; *Zeng et al., 2019*). As beige and brown adipocytes share similarities, it is possible that these factors may also affect sympathetic innervation in iWAT. Interestingly, *S100b*, but not *Tgfb1*, showed a regional pattern at the mRNA level during early iWAT development, with higher expression in the inguinal region in a PRDM16-dependent manner (*Figure 5—figure supplement 2F*). This result suggests that S100B may be one of the potential cues in regulating iWAT sympathetic development. It is also likely that sympathetic neurites are indirectly regulated by additional beige adipocyte-associated cell types, such as immune or stromal cells. Follow-up studies will need to evaluate the role of S100B and other potential factors in modulating sympathetic axon growth during early iWAT morphogenesis.

Our studies using an inducible adipocyte-specific *Prdm16* deletion model indicated a critical developmental window for the interactions between beige adipocytes and sympathetic nerve terminals. Restricted *Prdm16* deletion during early tissue morphogenesis resulted in a lasting reduction in parenchymal neurite density. However, *Prdm16* deletion in fully mature mice failed to alter sympathetic neurite density. These results indicate that sympathetic neurites in iWAT respond to signals from beige adipocytes or associated cells during early development. However, when the innervation pattern is fully established, such signals are no longer required for maintaining the innervation level during adulthood. In line with this, we have previously shown that cold-induced beige adipocytes do not promote sympathetic neurite outgrowth in adult iWAT when equivalent tissue regions are compared. Taken together, our data suggest that sympathetic neurite density is regulated by local cues from beige adipocytes during a specific developmental window and exhibits limited plasticity once the pattern is established.

Of note, a recent study demonstrated leptin-mediated central regulation of sympathetic innervation in adipose tissue (*Wang et al., 2020*). Specifically, chronic leptin treatment acting on hypothalamic neurons was found to rescue the defect in sympathetic innervation in iBAT and iWAT of adult leptin-deficient mice. It is possible that the central and local regulation of sympathetic innervation in adipose tissue acts with different timing and through distinct mechanisms. Interestingly, the sympathetic axon growth period we observed (P10-P21) largely overlaps with a postnatal leptin surge (P8-P20) reported previously (*Ahima et al., 1998*; *Wu et al., 2020a*). Future studies are needed to uncouple central (leptin surge) and local (beige adipocyte-associated factors) effects to fully understand how adipose sympathetic innervation is regulated by each mechanism. It is also worth noting that current studies rely on neurite morphological changes such as length to characterize sympathetic growth or remodeling. As adipocyte size dramatically changes in response to caloric excess or deprivation, sympathetic neurite density may appear different even without active remodeling. A better understanding of sympathetic neurite structural change will be assisted by identifying markers specific to actively remodeling neurites.

Thermogenic adipocytes have been demonstrated to provide metabolic benefits that may combat obesity and associated metabolic diseases. As thermogenic adipocytes are primarily induced by sympathetic stimulation, many studies have turned to the sympathetic nervous system in search of novel therapeutic targets for enhancing thermogenic adipocyte function. Our studies here demonstrated a critical developmental window during which beige adipocytes regulate sympathetic neurite density, providing fundamental knowledge about the development of the sympathetic nervous system in mouse subcutaneous white fat and providing a framework for future attempts to target this pathway for therapeutic benefit.

# Materials and methods

## Key resources table

| Reagent type (species) or resource | Designation | Source or reference | Identifiers | Additional information |
|---|---|---|---|---|
| Strain, strain background (*Mus musculus*) | *Adipoq-Cre* | Jackson Laboratory | RRID:IMSR_JAX:028020 | |
| Strain, strain background (*Mus musculus*) | *Adipoq-rtTA* | PMID:22451920 | RRID:IMSR_JAX:033448 | |
| Strain, strain background (*Mus musculus*) | *TRE-Cre* | Jackson Laboratory | RRID:IMSR_JAX:006234 | |
| Strain, strain background (*Mus musculus*) | *Prdm16*$^{lox/lox}$ | PMID:24439384 | RRID:IMSR_JAX:024992 | |
| Antibody | Anti-tyrosine hydroxylase (Rabbit polyclonal) | Millipore | Cat# AB152, RRID:AB_390204 | IF(1:200) |
| Antibody | Anti-tyrosine hydroxylase (Sheep polyclonal) | Millipore | Cat# AB1542, RRID:AB_90755 | IF(1:200) |
| Antibody | Anti-CD31/PECAM-1 (Goat polyclonal) | R and D Systems | Cat# AF3628, RRID:AB_2161028 | IF(1:200) |
| Antibody | Anti-UCP1 (Rabbit polyclonal) | Abcam | Cat# ab10983, RRID:AB_2241462 | IF(1:200) |
| Antibody | Anti-LYVE1 (Rabbit polyclonal) | Abcam | Cat# ab14917, RRID:AB_301509 | IF(1:200) |
| Antibody | Anti-PRDM16 (Sheep polyclonal) | R and D Systems | Cat# AF6295, RRID:AB_10717965 | WB(1:500) |
| Antibody | Anti-Lamin A/C (Mouse monoclonal) | Santa Cruz Biotechnology | Cat# sc-376248, RRID:AB_10991536 | WB(1:2000) |
| Antibody | Anti-Rabbit IgG (H+L), Alexa Fluor 568 (Donkey polyclonal) | Thermo Fisher Scientific | Cat# A10042, RRID:AB_2534017 | IF(1:200) |
| Antibody | Anti-Rabbit IgG (H+L), Alexa Fluor 647 (Donkey polyclonal) | Thermo Fisher Scientific | Cat# A32795, RRID:AB_2762835 | IF(1:200) |
| Antibody | Anti-Sheep IgG (H+L), Alexa Fluor 647 (Donkey polyclonal) | Thermo Fisher Scientific | Cat# A-21448, RRID:AB_2535865 | IF(1:200) |
| Antibody | Anti-Goat IgG (H+L), Alexa Fluor 568 (Donkey polyclonal) | Thermo Fisher Scientific | Cat# A-11057, RRID:AB_2534104 | IF(1:200) |
| Software, algorithm | Imaris | Bitplane | http://www.bitplane.com/imaris/imaris; RRID:SCR_007370 | |
| Software, algorithm | FilamentTracer | Bitplane | http://www.bitplane.com/imaris/filamenttracer; RRID:SCR_007366 | |

## Animals

Young wild-type mice of various ages were generated by crossing male and female mice from the C57BL/6J background (C57BL/6J, JAX 000664) obtained from the Jackson Laboratories and maintained in our facilities. The constitutive *Prdm16*$^{KO}$ (c*Prdm16*$^{KO}$) mice were generated as previously described (*Cohen et al., 2014*) by crossing *Adipoq-Cre* mice (JAX 028020) with *Prdm16*$^{lox/lox}$ mice. The inducible *Prdm16*$^{KO}$ (i*Prdm16*$^{KO}$) mice were generated by crossing *Adipoq-rtTA* (provided by Dr. Philipp E. Scherer) (*Sun et al., 2012*), *TRE-Cre* (B6.Cg-Tg(tetO-cre)1Jaw/J, JAX 006234), and *Prdm16*$^{lox/lox}$ mice. All animals in this study were male mice on a pure C57BL/6J background.

All mice were maintained on a 12 hr light/dark cycle with free access to food and water. To generate mice born and raised at thermoneutrality, pregnant female mice were housed at 30℃ 14 days

after vaginal plug formation until the pups reach the indicated ages. All other mice were housed at 23°C. For perinatal *Prdm16* deletion, pregnant female mice were fed with a chow diet containing 600 mg/kg doxycycline (Bio-Serv, S4107) 14 days after vaginal plug formation until the pups reach P21. For adult *Prdm16* deletion, the inducible *Prdm16*$^{KO}$ (Cre+ and Cre-) mice were placed on a doxycycline-containing chow diet for the indicated time. All other mice were fed with a standard rodent chow diet. For cold exposure experiments, mice were placed at 8°C for 48 hr with two mice in each cage. Animal care and experimentation were performed according to procedures approved by the Institutional Animal Care and Use Committee at the Rockefeller University.

## iWAT regional dissection

Various regions of iWAT were dissected for qPCR or western blot analyses as illustrated in *Figure 1A* and *Figure 1—figure supplement 2K*. After removal of the lymph node, the region between the bottom two dotted lines, guided by the entry of the main blood vessel in the inguinal portion and the upper boundary of the lymph node, was dissected as the inguinal region. The region from the upper boundary of the lymph node to the back was considered as the dorsolumbar region. When indicated, the dorsolumbar region was further divided into dorsomedial and dorsolateral regions by making a cut alongside the blood vessel that travel through the dorsolumbar region.

## Gene expression analysis

Total RNA was extracted from tissue using TRIzol (Invitrogen) along with RNeasy kits (QIAGEN). An RNeasy mini kit was used for adult tissue samples, while an RNeasy micro kit was used for small tissue samples from young mice. For qPCR analysis, RNA was reverse transcribed using the high-capacity cDNA reverse transcription kit (Applied Biosystems). cDNA was used in qPCR reactions containing SYBR-green fluorescent dye (Applied Biosystems). Relative mRNA expression was determined by normalization with *Tbp* (TATA-box binding protein) levels using the ΔΔCt method. The sequences of primers used in this study are listed in *Supplementary file 1*.

## Nuclear extraction and immunoblotting

Frozen iWAT and iBAT were minced and homogenized in a hypotonic buffer (10 mM HEPES, 10 mM KCl, 1.5 mM MgCl$_2$, 0.5 mM DTT, and 1x protease inhibitor cocktail (cOmplete Mini, Roche)) by a dounce homogenizer. Homogenate was incubated on ice for 10 min and then mixed with 1/20 vol of 10% IGEPAL CA-630 (Sigma-Aldrich, I8896). Samples were then filtered through a 100 μm cell strainer and centrifuged at 1000 x g for 10 min. After centrifugation, lipid and cytoplasmic fractions were removed and nuclear pellets were resuspended in lysis buffer (20 mM HEPES, 1.5 mM MgCl$_2$, 0.42 M NaCl, 0.2 mM EDTA, 0.5 mM DTT, 1x protease inhibitor cocktail, and 20% Glycerol). Samples were incubated on ice for 30 min and vortexed for 15 s every 10 min during the incubation. After lysis, samples were centrifuged at 20,000 x g for 10 min and the supernatant was taken as the nuclear extract. The following antibodies were used in immunoblotting: anti-PRDM16 (1:500, R and D systems, AF6295), anti-Lamin A/C (1:2000, Santa Cruz, sc-376248).

## Adipo-Clear

Adipo-Clear was performed following a previously published protocol (*Chi et al., 2018a*; *Chi et al., 2018b*). In this study, primary antibodies including anti-UCP1 (1:200, abcam, ab10983), anti-TH (1:200, Millipore, AB1542 and AB152), anti-CD31 (1:200, R and D systems, AF3628), and anti-LYVE1 (1:200, abcam, ab14917), as well as secondary antibodies conjugated with Alexa-568 and Alexa-647 (1:200, Thermo Fisher Scientific, A10042, A21448, A32795, A11057) were used.

Immunostaining and imaging with iBAT cryo-sections were dissected from mice perfused and fixed with 1x PBS followed by 4% PFA. Harvested iBAT samples were post-fixed in 4% PFA at 4°C overnight and subsequently washed with 1x PBS for 1 hr at RT three times. Samples were then delipidated and permeabilized as described in the Adipo-Clear protocol (*Chi et al., 2018a*). Fully delipidated samples were incubated in 25% sucrose/PBS solution for 2 hr until sinking, and then frozen in Tissue-Tek O.C.T Compound (Sakura Finetek USA, 4583). Frozen iBAT samples were sectioned into 40 μm slices using a Leica CM3050 S cryostat. Cryo-sections were blocked with PBS/0.1% Triton X-100/0.05% Tween 20/2 μg/ml heparin (PtxwH buffer) containing 3% donkey serum for 1 hr at RT, and then incubated with primary antibodies diluted in PtxwH buffer at RT overnight. Samples were

then rinsed in PtxwH buffer for 5 min, 10 min, and 30 min to remove unbound antibodies. Secondary antibodies diluted in PtxwH buffer were then applied to samples at RT for 4 hr. Samples were next rinsed with PtxwH buffer for 5 min, 10 min, and 30 min, followed by 1x PBS for 10 min twice. Finally, samples were immersed in antifade mountant (ProLong Gold, ThermoFisher Scientific, P10144) and sealed with a coverslip. Anti-TH (1:200, Millipore, AB152) and Alexa-647 conjugated anti-rabbit secondary (1:200, Invitrogen, A32795) antibodies were used for staining cryo-sections. Fluorescently labeled samples were imaged on an inverted LSM 880 NLO laser scanning confocal and multiphoton microscope (Zeiss) with a 20X lens (NA 0.8).

## Light sheet microscopy

Whole-tissue iWAT samples were all imaged on a light sheet microscope (Ultramiscroscope II, LaVision Biotec) equipped with 1.3X and 4X objective lenses and an sCMOs camera (Andor Neo). Images were acquired with the ImspectorPro software (LaVision BioTec). Samples were positioned in an imaging chamber filled with benzyl ether and illuminated from one side by the laser light sheet with 488, 561, and 640 nm laser channels. Samples were scanned at a step-size of 4 µm for the 1.3x objective and 3 µm for the 4x objective.

## Image processing

All images and videos were generated using Imaris software (version 9.5.1, Bitplane). 3D tissue reconstruction was generated using the 'Volume' function. Maximum intensity projections and optical slices were obtained using the 'Ortho Slicer' function. All images were captured using the 'Snapshot' tool, while all videos were made using the 'Animation' tool.

## Neurite density quantification

Inguinal regions of iWAT from iKO and control iWAT samples were imaged with the 4X objective lens on the light sheet microscope. Three animals from each group were imaged and analyzed. In each 3D image, we randomly isolated small cuboidal volumes (4–8 volumes per sample) that were completely contained within lobules using the 'Surfaces' tool followed by the mask channel option of Imaris. Volumes of the isolated segments were automatically generated by 'Surfaces'. To sample parenchymal neurites, we avoided placing volumes in areas that contain nerve bundles or blood vessel innervation. Using the 'Filament' tool, we computationally reconstructed parenchymal neurites by automatically tracing the TH signal and calculated the total neurite length within each volume. We presented the ratio of total neurite length (mm) by regional volume ($mm^3$) as neurite density within a volume. To adjust for adipocyte size/number, we manually counted adipocyte number as shown by the tissue autofluorescence signal from multiple representative slices within each volume. The average adipocyte number per slice was then multiplied by the height (z depth) of that volume to generate a factor representing adipocyte density. The ratio of total neurite length (mm) by adipocyte density (arbitrary unit) is presented (*Figure 4—figure supplement 2*).

## Statistical analysis

All statistical analyses were performed using GraphPad Prism 8 (GraphPad Software, San Diego, CA, USA). For gene expression analysis, neurite density quantification, and body weight measurement, we estimated the approximate effect size based on independent preliminary studies. When indicated, an unpaired two-tailed Student's t test was used to analyze statistical differences. Two-way ANOVA followed by Bonferroni's multiple comparisons test was applied to determine the statistical differences for the rest of the analyses. The statistical details for each experiment can be found in the figure legends. p Values below 0.05 were considered significant throughout the study.

## Acknowledgements

We thank Dr. Philipp Scherer for the *Adipoq-rtTA* mice line. We thank the Cohen lab for critical comments. We also thank Christina Pyrgaki, Tao Tong, Katarzyna Cialowicz, and Alison North from the Rockefeller Bioimaging Resource Center for assistance and support. JC is supported by the Center for Basic and Translational Research on Disorders of the Digestive System through the generosity of the Leona M and Harry B Helmsley Charitable Trust. This work was supported by the National

Institute of Diabetes and Digestive and Kidney Disease grant R01 DK120649 and by the American Diabetes Association Pathway Program (Grant # 1–17-ACE-17).

## Additional information

### Funding

| Funder | Grant reference number | Author |
| --- | --- | --- |
| Leona M. and Harry B. Helmsley Charitable Trust | Center for Basic and Translational Research on Disorders of the Digestive System Pilot Award | Jingyi Chi |
| National Institute of Diabetes and Digestive and Kidney Diseases | R01 DK120649 | Paul Cohen |
| American Diabetes Association | Grant # 1-17-ACE-17 | Paul Cohen |

The funders had no role in study design, data collection and interpretation, or the decision to submit the work for publication.

### Author contributions
Jingyi Chi, Conceptualization, Data curation, Formal analysis, Funding acquisition, Validation, Investigation, Visualization, Methodology, Writing - original draft, Project administration, Writing - review and editing; Zeran Lin, William Barr, Formal analysis, Investigation, Writing - review and editing; Audrey Crane, Investigation, Writing - review and editing; Xiphias Ge Zhu, Investigation, Visualization, Writing - review and editing; Paul Cohen, Conceptualization, Supervision, Funding acquisition, Project administration, Writing - review and editing

### Author ORCIDs
Jingyi Chi (iD) https://orcid.org/0000-0001-6013-8544
Zeran Lin (iD) https://orcid.org/0000-0003-4418-2443
William Barr (iD) https://orcid.org/0000-0001-6723-7684
Audrey Crane (iD) https://orcid.org/0000-0003-1573-6099
Paul Cohen (iD) https://orcid.org/0000-0002-2786-8585

### Ethics
Animal experimentation: This study was performed in strict accordance with the recommendations in the Guide for the Care and Use of Laboratory Animals of the National Institutes of Health. All of the animals were handled according to approved institutional animal care and use committee (IACUC) protocols (#18016-H) of The Rockefeller University.

### Decision letter and Author response
Decision letter https://doi.org/10.7554/eLife.64693.sa1
Author response https://doi.org/10.7554/eLife.64693.sa2

## Additional files

### Supplementary files
• Supplementary file 1. qPCR primer sequences used in this study.

• Transparent reporting form

### Data availability
All data generated or analyzed during this study are included in the manuscript and supporting files.

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
