## [Decision Letter]

**Acceptance summary:**

This paper provides a detailed description of the development and patterning of murine inguinal white adipose tissue. This study provides key information regarding the timing of adipose innervation in relation to beige adipogenesis. The finding that perinatal beige adipogenesis, which is driven by the primary beige adipogenic transcription factor PRDM16, is critical for establishing the network of parenchymal neurites present in adult mice is of particular interest.

**Decision letter after peer review:**

Thank you for submitting your article "Early postnatal interactions between beige adipocytes and sympathetic neurites regulate innervation of subcutaneous fat" for consideration by *eLife*. Your article has been reviewed by three peer reviewers, one of whom is a member of our Board of Reviewing Editors, and the evaluation has been overseen by Richard White as the Senior Editor. The reviewers have opted to remain anonymous.

The reviewers have discussed the reviews with one another and the Reviewing Editor has drafted this decision to help you prepare a revised submission.

Summary:

In this article, Chi et al. provide a detailed description of the development and patterning of murine inguinal white adipose tissue. This area of study, while of critical importance in understanding the dynamics of adipose tissue innervation and systemic metabolism, has suffered from a dearth of experimental evidence despite the ever-increasing interest in adipose thermogenesis being adapted for therapeutic purposes in humans. This study provides key information regarding the timing of adipose innervation in relation to beige adipogenesis, as well as careful quantification of neurite densities with appropriate normalization strategies. These data will be informative for future studies involving sympathetic regulation of adipose function. The finding that perinatal beige adipogenesis, which is driven by the primary beige adipogenic transcription factor PRDM16, is critical for establishing the network of parenchymal neurites present in adult mice is of particular interest as it further suggests a relatively fixed sympathetic tone in adult mice.

Essential revisions:

1) Strictly speaking, *Prdm16* is inactivated in all adipocytes (Adipoq+); however, the claim is that beige adipocytes drive the innervation. A fairly convincing case is made that it's the beige adipocytes, but can the authors rule out that white adipocyte *Prdm16* is regulating this process. Is *Prdm16* only expressed in UCP1+ adipocytes of the iWAT depot? This should be discussed/addressed.

2) The Spiegelman group has recently reported that TGFβ1 and *S100b* are determinants of sympathetic innervation in adipose tissue. Are these factors involved in the early development of the innervation patterning? It is not essential that new neurotrophic factor be unveiled in this study, but consideration should be made for the known regulators.

3) Prior papers have suggested a more dynamically regulated sympathetic arborization network within the adult mouse iWAT, such as via a dramatic induction of innervation following chronic leptin administration in adult Ob/Ob mice (PMID: 28918935). Additionally, another study on iWAT innervation showed a dramatic decrease in innervation following diet-induced obesity by a high-fat diet (PMID: 32699414). Given the relatively fixed patterning that the authors have shown is established in the critical window of development immediately following birth, do the authors believe that these studies present contradictory findings regarding the plasticity of adult iWAT innervation?

4) Can the compromised innervation in perinatal *Prdm16*^KO^ mice be naturally rescued/compensated for as animals age? Or is the defect in innervation permanent? The answer may speak to whether the neonatal period is the only critical time window to fully establish iWAT thermogenic capacity. Data on this point would be helpful but is not strictly required.

5) The authors describe placing the early postnatal mice at thermoneutrality (30 ^o^C) in order to observe the development of the sympathetic arborizations independent of the minor cold stress of room temperature. However, given the fact that these pups lack the same amount of fur as adult mice, the true thermoneutral point for these mice may be notably higher than 30 ^o^C, and thus some cold stress still exists and the conclusion that this increase in innervation is a developmentally hard-wired program may not be fully substantiated. The authors should confirm the thermoneutral point for these pups by observing the relationship between pup body temperature and ambient temperature, and, if necessary, reconduct the experiment at a higher ambient temperature.

6) In Figure 2 it appears that the perinatal emergence or UCP1 positive beige adipocytes staining in WT animals is conspicuously denser around the central lymph node (This area seems to correspond with what is referred to as the "core" in the text). Is this an accidental observation or is there reason to believe that innervation or vascularization around the lymph node or lymph vessels may provide important trophic signals to drive PDRM16 , UCP1, CD36 and TH innnervation? Have the authors examined any lymph vessel specific markers or approaches could be targeted to the WAT lymph node or its supply to directly challenge this suggestion?

7) On a related note, Figure 2—figure supplement 1: The authors remark on the apparent reduction in relative UCP1 mRNA levels in inguinal adipose tissue after p14 (and to some degree PRDM16, yet NOT relative mRNA levels for most other adipose beiging-related markers). Based on the apparent specialization of UCP1 density around the core (near the lymph node) and diminished density as one radiates outward form the core, this peri-lymph node region may be quantitatively different and more dense compared to other regions farther away. Thus, the present relative UCP1 and PRDM16 mRNA level statistics at days greater that p14 may be artificially low, depending on exactly how the authors sample or average over the entire depot post day 14. This may be in part because the depot itself grows and its extremities extend farther and farther away from the central lymph node at days after p 14, so there might be an artificial "diluting " of relative UCP1 mRNA and PRDM16 reported.

---

## [Author Response]

Essential revisions:1) Strictly speaking, Prdm16 is inactivated in all adipocytes (Adipoq+); however, the claim is that beige adipocytes drive the innervation. A fairly convincing case is made that it's the beige adipocytes, but can the authors rule out that white adipocyte Prdm16 is regulating this process. Is Prdm16 only expressed in UCP1+ adipocytes of the iWAT depot? This should be discussed/addressed.

Thank you for raising this important point. While this question would be best answered by directly visualizing the localization of PRDM16-expressing adipocytes in iWAT, we could not find any PRDM16 antibody suitable for immunofluorescence or immunohistochemistry in adipose tissue. Therefore, as an alternate approach, we made use of the regionality in iWAT to isolate predominantly beige vs. white regions from this depot and analyzed *Prdm16* mRNA and protein levels in these regions (Figure 3—figure supplement 2I and J). As shown in our current and previous work (Chi et al., 2018), the inguinal region is enriched with early postnatal and cold-induced beige adipocytes in adults, whereas the dorsolateral region is mostly devoid of early postnatal beige adipocytes and remains UCP1-negative even following prolonged cold exposure in adulthood. These observations suggest that the inguinal and dorsolateral regions can be considered as beige and white regions of iWAT, respectively.

In Figure 3—figure supplement 2I, we performed qPCR on the two regions isolated from control and c*Prdm16*^KO^ mice at postnatal day 14. A detailed description about regional dissection is included in the Materials and methods section. The control dorsolateral region showed significantly lower expression of *Prdm16* mRNA than the control inguinal region. Importantly, *Prdm16* mRNA levels in the control dorsolateral region were indistinguishable from those in c*Prdm16*^KO^ dorsolateral or inguinal regions, suggesting that the wild-type dorsolateral region naturally expresses very low levels of *Prdm16* mRNA with levels indistinguishable from *Prdm16* knockout samples.

In Figure 3—figure supplement 2J, we further assessed PRDM16 protein levels across multiple fat depots of adult mice. Consistently, the dorsolateral region exhibited a considerably lower level of PRDM16 compared with the inguinal region in wild-type iWAT, while there was no detectable PRDM16 in the iWAT of *cPrdm16^KO^* mice or wild-type eWAT. Although there was still a minimal level of PRDM16 protein in the dorsolateral region of iWAT, this can be attributed to the small number of beige adipocytes in the dorsolateral region. Altogether, with the *Prdm16* mRNA and protein assessments in the different regions of iWAT, we believe that *Prdm16* is minimally expressed in white adipocytes in iWAT, and thus feel that its deletion in white adipocytes contributes minimally to the changes in sympathetic innervation.

2) The Spiegelman group has recently reported that TGFβ1 and S100b are determinants of sympathetic innervation in adipose tissue. Are these factors involved in the early development of the innervation patterning? It is not essential that new neurotrophic factor be unveiled in this study, but consideration should be made for the known regulators.

In the publications you allude to above, the roles of *S100b* and *Tgfb1* in regulating sympathetic innervation were mainly investigated in iBAT. Since beige and brown adipocytes share similarities, it is possible that these factors may also regulate sympathetic innervation of beige adipocytes in iWAT. Based on our observations, potential candidates that regulate the growth of sympathetic neurites in iWAT during early development may exhibit a PRDM16-dependent regional pattern. We performed qPCR to assess the mRNA levels of *S100b* and *Tgfb1* in the inguinal and dorsolateral regions isolated from control and c*Prdm16*^KO^ mice at postnatal day 14. Interestingly, *S100b*, but not *Tgfb1*, is enriched in the control inguinal region in a PRDM16-dependent manner (Figure 5—figure supplement 2F), suggesting *S100b* may play a role in regulating sympathetic growth in beige adipocytes. Additional experiments are required to examine the function of *S100b* and *Tgfb1* in sympathetic development during early iWAT morphogenesis. Importantly, sympathetic axonal growth has been shown to be regulated by multiple factors in other organ systems, such as heart (Kimura et al., 2012). It is likely that follow-up mechanistic studies can uncover additional proteins important for sympathetic innervation establishment in iWAT.

3) Prior papers have suggested a more dynamically regulated sympathetic arborization network within the adult mouse iWAT, such as via a dramatic induction of innervation following chronic leptin administration in adult Ob/Ob mice (PMID: 28918935). Additionally, another study on iWAT innervation showed a dramatic decrease in innervation following diet-induced obesity by a high-fat diet (PMID: 32699414). Given the relatively fixed patterning that the authors have shown is established in the critical window of development immediately following birth, do the authors believe that these studies present contradictory findings regarding the plasticity of adult iWAT innervation?

Thank you for raising this important point. Our current studies focus on sympathetic development in iWAT. Our results support that early sympathetic axon growth is guided by beige adipocyte-associated molecules, and that such molecules are not responsible for maintaining sympathetic innervation of iWAT outside the critical developmental window. On the other hand, the reduced sympathetic neurite density following high-fat diet feeding may be due to a form of axon degeneration. Axon growth and degeneration are distinct phenomena that usually occur through different mechanisms. While axon growth is regulated by neurotrophic and axon guidance molecules (Glebova and Ginty, 2005), axon degeneration may be caused by metabolic alterations or mechanical injury (Coleman, 2005). It will be interesting for future studies to examine mechanisms and physiological consequences of reduced sympathetic neurite density in adipose tissue during the progression of obesity.

It has recently been reported that leptin deficient mice also show dramatically reduced sympathetic neurite density in iBAT and iWAT, while chronic leptin treatment stimulates adipose sympathetic axon growth by acting on neurons in the hypothalamus in adult mice (Wang et al., 2020). These results unveiled an important central (or top-down) pathway that regulates sympathetic neuron properties, which is possibly acting upon a different time frame than target tissue-derived cues. Additional studies are needed to unravel the contribution of central vs. local regulation of adipose tissue sympathetic innervation. Taken together, we believe that sympathetic axon degeneration in diet-induced obesity and leptin-stimulated axonal regrowth in ob/ob mice are distinct phenomena, and therefore not necessarily contradictory to our findings on the critical time window for local/target tissue-derived regulation.

4) Can the compromised innervation in perinatal Prdm16^KO^ mice be naturally rescued/compensated for as animals age? Or is the defect in innervation permanent? The answer may speak to whether the neonatal period is the only critical time window to fully establish iWAT thermogenic capacity. Data on this point would be helpful but is not strictly required.

We agree with the reviewers that this is an important point. To assess potential rescue of the innervation defect following perinatal deletion of *Prdm16*, we performed the same perinatal deletion of *Prdm16* (E14 to P21) but let animals reach 8 weeks of age on a chow diet, hereafter referred to as i*Prdm16*^KO^ (perinatal, 8wk) (Figure 4—figure supplement 4A). As iWAT continues to expand in young adult mice (from 5-week-old to 8-week-old), we reasoned that newly developed adipocytes during iWAT expansion might also affect the sympathetic innervation pattern. Consistent with the results from the i*Prdm16*^KO^ (perinatal, 5wk) experiment, we observed significantly reduced expression of *Prdm16* and other thermogenic genes at the mRNA level in i*Prdm16*^KO^ (perinatal, 8wk) relative to control (Figure 4—figure supplement 4B-D). Importantly, TH^+^ fibers appeared substantially sparser in i*Prdm16*^KO^ (perinatal, 8wk) than control. These data demonstrate that the innervation defect resulting from perinatal ablation of *Prdm16* persists during a natural period when iWAT continues to actively grow. However, it is possible that sympathetic innervation may grow back as mice age further. Future studies will be necessary to examine this biology in aged mice.

5) The authors describe placing the early postnatal mice at thermoneutrality (30 ^o^C) in order to observe the development of the sympathetic arborizations independent of the minor cold stress of room temperature. However, given the fact that these pups lack the same amount of fur as adult mice, the true thermoneutral point for these mice may be notably higher than 30 ^o^C, and thus some cold stress still exists and the conclusion that this increase in innervation is a developmentally hard-wired program may not be fully substantiated. The authors should confirm the thermoneutral point for these pups by observing the relationship between pup body temperature and ambient temperature, and, if necessary, reconduct the experiment at a higher ambient temperature.

We apologize for the confusion. It is not our intention to argue that the development of sympathetic innervation is independent of cold stress with this thermoneutral experiment. We only wish to use these data to support the argument that emergence of early postnatal beige adipocytes may be spontaneous rather than induced by sympathetic stimulation. This same phenomenon has been examined in Wu et al., 2020. In this study, mice with genetic denervation of the sympathetic nervous system developed early postnatal UCP1+ adipocytes similar to the control group, supporting that sympathetic activation is not required to recruit these cells. We modified this section in the revised manuscript to make this point more clearly.

6) In Figure 2 it appears that the perinatal emergence or UCP1 positive beige adipocytes staining in WT animals is conspicuously denser around the central lymph node (This area seems to correspond with what is referred to as the "core" in the text). Is this an accidental observation or is there reason to believe that innervation or vascularization around the lymph node or lymph vessels may provide important trophic signals to drive PDRM16 , UCP1, CD36 and TH innnervation? Have the authors examined any lymph vessel specific markers or approaches could be targeted to the WAT lymph node or its supply to directly challenge this suggestion?

We agree with the reviewers that the “core” region of iWAT, which is close to the inguinal lymph node and the main blood vessel that travels through the inguinal portion of iWAT, may contain important cell types or trophic factors that promote beige adipocyte recruitment and the subsequent establishment of sympathetic innervation. To visualize lymphatic vessels in iWAT, we performed imaging using an antibody targeting lymphatic vessel endothelial hyaluronan receptor 1 (LYVE1), a well-recognized marker for lymphatic endothelial cells in various tissue types (Podgrabinska et al., 2002; Potente and Mäkinen, 2017). In addition to the lymph node, we observed that LYVE1 staining labeled cell-like and vessel-like structures in iWAT (Figure 5—figure supplement 2A-E). The LYVE1+ vessel originating from the lymph node is likely a lymph vessel. Importantly, this vessel lies adjacent to the main blood vessel that is heavily innervated by TH^+^ fibers and travels through the inguinal region (Figure 5—figure supplement 2D-E). Although we agree that blood and lymph vessels in this region may elaborate important factors to drive iWAT patterning, our current imaging data can neither support nor refute the reviewer’s question on causality. Further experiments that manipulate lymph vessels and associated cell types within the adipose tissue will be required to address this question.

7) On a related note, Figure 2—figure supplement 1: The authors remark on the apparent reduction in relative UCP1 mRNA levels in inguinal adipose tissue after p14 (and to some degree PRDM16, yet NOT relative mRNA levels for most other adipose beiging-related markers). Based on the apparent specialization of UCP1 density around the core (near the lymph node) and diminished density as one radiates outward form the core, this peri-lymph node region may be quantitatively different and more dense compared to other regions farther away. Thus, the present relative UCP1 and PRDM16 mRNA level statistics at days greater that p14 may be artificially low, depending on exactly how the authors sample or average over the entire depot post day 14. This may be in part because the depot itself grows and its extremities extend farther and farther away from the central lymph node at days after p 14, so there might be an artificial "diluting " of relative UCP1 mRNA and PRDM16 reported.

We thank the reviewer for this point. We have now included a detailed description of our dissection method and additional figures to illustrate depot boundaries or “extremities”. In Figure 2—figure supplement 1F-H, we presented images containing UCP1 and tissue autofluorescence signals that correspond to iWAT samples of P14, P21, and P28 mice. The tissue autofluorescence signals outline tissue boundaries following dissection. Among the three time points, the proportion of UCP1 signal within the tissue boundary is the lowest at P14. This suggests that qPCR results of *Ucp1* at P14 may be an underestimation and argues against a dilution of *Ucp1* signal at later timepoints. However, quantitative mRNA or protein analysis of bulk tissue pieces cannot directly address the changes of *Ucp1* levels in early postnatal beige adipocytes over time. Future studies may involve single cell or single nuclei RNAseq to track *Ucp1* content within individual cells over the developmental time course.

References:

Coleman M. 2005. Axon degeneration mechanisms: commonality amid diversity. Nat Rev Neurosci 6:889–898. doi:10.1038/nrn1788Kimura K, Ieda M, Fukuda K. 2012. Development, Maturation, and Transdifferentiation of Cardiac Sympathetic Nerves. Circ Res 110:325–336. doi:10.1161/CIRCRESAHA.111.257253Podgrabinska S, Braun P, Velasco P, Kloos B, Pepper MS, Jackson DG, Skobe M. 2002. Molecular characterization of lymphatic endothelial cells. Proc Natl Acad Sci 99:16069–16074. doi:10.1073/pnas.242401399Potente M, Mäkinen T. 2017. Vascular heterogeneity and specialization in development and disease. Nat Rev Mol Cell Biol 18:477–494. doi:10.1038/nrm.2017.36